# Role of Hormones and the Potential Impact of Multiple Stresses on Infertility

**Shanmugam Ramya [1], Prasad Poornima [1], Arumugam Jananisri [1], Irudhayaraj Peatrise Geofferina [1], Venkataramanaravi Bavyataa [1], Murugan Divya [1], Palanisamy Priyanga [1], Jeganathan Vadivukarasi [1], Senthil Sujitha [1], Selvarasu Elamathi [1], Arumugam Vijaya Anand [1,*] and Balasubramanian Balamuralikrishnan [2]**

[1]  Department of Human Genetics and Molecular Biology, Bharathiar University, Coimbatore 641046, Tamil Nadu, India; ramyashanmugam727@gmail.com (S.R.)
[2]  Department of Food Science and Biotechnology, College of Life Science, Sejong University, Seoul 05006, Republic of Korea
*   Correspondence: avahgmb@buc.edu.in

**Abstract:** Infertility has a remarkable global impact affecting approximately about 48 million couples worldwide. One of the most contended concerns in reproductive biology is the prospective influence of psychological stress on pregnancy rates. Individuals struggling to conceive face a stupendous amount of emotional turbulence and have a greater risk for psychological vulnerability. Both stress and infertility reinforce the impact of each other; hence, there exists a reciprocal relationship. Stress may be the major contributor to subsequent infertility. Infertility treatments may enhance stress levels as well as reduce treatment outcomes. The biological mechanisms that interlink stress and infertility are the outcome of the hormonal actions at the brain level, where they stimulate or suppress the hypothalamic-pituitary-adrenal axis (HPA) and have a potential influence on the secretion of the respective hormone by the reproductive organs and the pregnancy outcomes. Sex hormones play an essential role in reproductive biology as well as in general physiology where they generate the cycle and provide a potential environment for pregnancy. This article reviews the impact of stress on reproductive rates and the implications of sex hormones on infertility. Additionally, it suggests strategies to overcome the stress conditions and the scenarios that may lead to stress.

**Keywords:** infertility; stress; sex hormones; hypothalamic-pituitary-adrenal axis; pregnancy stress management



## 1. Introduction

Stress is an inevitable outcome of modern life which is expressed as an uncomfortable condition of mental and physiological arousal that people experience when they perceive a harmful or dangerous situation to their well-being [1]. With continuously changing lifestyles and social systems, the ability to deal with stress and limit its consequences is essential for living a balanced and healthy life [2]. The basis of the complexion, duration, and severity of stress responses can affect homeostasis and have various impacts on the body, including life-threatening effects and even death [3].

Infertility is one of the global health problems and the most distressing condition affecting about 186 million people [4] that may carry significant psychological trauma. Personal stress, pressure from marital relationships, and social networks for married couples' further family functioning and educational level are negatively correlated with infertility [5]. Although not fatal, it has been described as a serious life-changing problem where one is unable to establish clinical pregnancy [6]. Behind the workup of infertility, traditionally, the female factor is considered the major driving force, and the malefactor is predicted to be 50% of the cause, with a peculiar contribution in 20% of instances [7,8].

In males and females, the influence of mental disorders is often considered the reason for idiopathic (unexplained) infertility including stress, depression, sleep, and eating disorders.

The knowledge of the relationship between stress and infertility is still not broadly accepted. However, it has been reported that stress is a common factor in the development of unexplained reproductive disorders [9]. The major drawback seems to be the hardship in administering the correlation between emotional factors and reproductive outcomes [4]. The inability to reproduce naturally can cause feelings of shame, guilt, and low self-esteem, thus increasing their psychological vulnerability, and these negative feelings may lead to varying degrees of depression, anxiety, distress, and a poor quality of life. This review focuses on elucidating the knowledge of the impact of stress on infertility or reproductive outcomes and the role of hormones in causing stress as well as affecting the fertility rate, which also includes various stress-related ailments that lead to infertility. International databases (PubMed, Scopus, Web of Science) and Google Scholar were searched for articles published from 2000 to February 2023. The search procedure was performed in English using keywords such as "stress", "depression", "infertility", and "reproductive hormones". The articles were evaluated in terms of their titles, abstracts, and full texts.

## 2. Possible Mediators of Infertility

Stress is always manifested as a phenomenon that fails to attain the genetic potential and to cope with the environment of any individual [10]. Indelible diagnosis and therapeutic methods may have a pessimistic impact on sexual lives and can also induce stress among infertile couples [4]. Acute stress and chronic stress are two distinct kinds of stress. Acute stress is generally present in times of emergency, such as fighting. The changes that occur in the structure and function of specific chemicals in the brain trigger the emotional cognitive system, leading us to make stress-coping judgments. As a result, acute stress is frequently beneficial to the body, and it also enhances resistance to infection and minimizes the ability of an individual to produce a powerful immunological reflex along with the subsequent rise in ailments [11,12]. However, chronic stress is extensively involved in the pathogenesis of various diseases, and it is now thought to include both professional stress and atypical adversity. It may cause sleeplessness, gastrointestinal issues, anxiety, and depression, as well as an increased risk of complications associated with cardiac, cancer, and mental illness in addition to adversely affecting fertility. It also hampers the process of immune cell migration to the site of inflammation or infection [4,11]. The primary cause of infertility is psychological stress, and it is often preventable. The other mediators reported are metabolic stress and oxidative stress, which are interlinked with psychological stress [13].

### 2.1. Psychological Stress

Psychological stress is common among infertile women, and it is often manifested as chronic stress. Untreated stress can interfere with the pregnancy success rate or persistence with fertility treatment. The psychological stress that leads to a cytokine storm induces, in turn, oxidative stress which may trigger disruptions in the blood–testis barrier and lead to abnormal semen characteristics [14]. It causes a delay in attempting fertility that has a negative impact on the couple's fecundity. The disputing psychological stress is reported as the major reason for severe reductions in testicular function leading to infertility [15]. Moreover, psychological stress is shown to be associated with high levels of female infertility that establish greater levels of undesirable pregnancy which includes repeated implantation failure and miscarriage. Psychological distress has a high probability of directly causing endometrial dysfunction by disrupting the secretion and regulation of endometrial hormones [16]. Prolonged psychological distress leads to a higher rate of discontinuous medically assisted reproduction. The vital environmental factor which adversely influences fertility is psychological stress which is associated with multiple health consequences and is considered to be a chronic experience of stress causing infertility [17].

## 2.2. Emotional Stress

Emotional stress is another major consequence of infertility, and it is broadly accepted [18]. Many men and women experience emotional suffering as a result of failure to conceive. Various reports establish a relationship manifested as a vicious cycle where infertility results in emotional stress, and that can promptly impact fertility by modifying hypothalamic-pituitary pathways, or it may cause tubal spasm dyspareunia, decreased male libido, and frigidity [19,20]. Despite the lack of biological reports regarding the biological relationship between emotional stress and infertility, rigorous care should be taken. Maternal stress is widely blamed for infertility due to miscarriages, late pregnancy problems, and decreased fetal development. Reducing the burden of infertility on women would not only stress the overall stress but also aid in no biologic ways to improving pregnancy chances [18].

## 2.3. Metabolic Stress

In recent years, the assessment of metabolic risk factors is considered highly essential for understanding reproductive disorders, especially since it may contribute in figuring out the cause behind idiopathic infertility, which might help in enhancing fertility treatment outcomes. Metabolic stress is an underlying cause of various metabolic disorders including obesity, osteoporosis, amenorrhea, or athlete triad (an eating disorder that leads to being underweight) [21]. The imbalance between reproduction and metabolism is mainly due to PCOS which often leads to ovarian dysfunction; further, it causes metabolic syndrome in females. Obesity can contribute to and cause permanent female infertility, and a higher BMI causes a potential reduction in serum adiponectin levels and elevation of insulin, which triggers hyperandrogenemia, inducing the risk of menstrual irregularity, PCOS, acne, and hirsutism. Women with greater BMI exhibit lower fecundity despite many ART treatment cycles [22,23]. The female with athlete triad experiences menstrual dysfunction which is accompanied by the risk of infertility [24].

## 2.4. Oxidative Stress

An imbalance in Oxidative stress may cause a serious effect on female reproduction and female reproductive ailments such as PCOS, endometriosis, preeclampsia, etc. [25]. The disproportion in equilibrium between reactive oxygen species (ROS) and antioxidant causes peroxidative damage to sperm motility due to scrotal heat, metabolite reflux, and testicular hypoxia. Oxidative stress is also caused by various metals, organic solvents, and ionizing radiation [26]. Oxidative stress can cause or worsen obesity, insulin resistance, vitamin D deficiency, and immune dyscrasia in PCOS physiopathology which is a remarkable risk factor for infertility [27]. Several cellular mechanisms are regulated by oxidative stress. The expression and secretion of sex hormone-binding globulin are inhibited by oxidative stress, by down regulating the HNF-4$\alpha$ in vitro, and this may be an essential factor in promoting hyperandrogenemia in PCOS. Abnormal levels of oxidative stress are found in women with PCOS that are independent of being overweight [28]. The oxidative stress is indicative of the functional efficacy inflammation markers such as resting level of seminal interleukin in men with impaired fertility [29].

## 2.5. Preconception Stress

Preconception stress is common among both men and women of reproductive age, and it may affect fertility, pregnancy, and neonatal outcomes. Every year, miscarriage pregnancies are accounted for about 23 million globally, resulting in 44 pregnancy losses each minute [30,31]. Preconception stress can be due to marital conflict, difficult bereavement, or any other psychiatric disorders [32]. Routine screening of the paternal parent's mental health is essential to prevent the impact of the maternal parent's depression on the child. Stress can disrupt the functions of the HPG axis which results in a reduction in the process of steroidogenesis and spermiogenesis, which may also provoke the chance of infection and inflammation and worsen semen quality [33]. Maternal stress is highly responsible for

adverse pregnancy outcomes such as preterm birth, miscarriages, minimum development for gestational age, and low birth weight [34].

## 3. Reproductive Hormonal Changes during Stress and Infertility Conditions

Hormones administer and coordinate human sexual development, sexuality, reproduction, developmental plasticity, or phenotypic rearrangements that happen perpetually at the entire ontogeny including follicle formation and ovulation which involves hormones such as LH and Follicle-Stimulating Hormone (FSH), as their functions are illustrated in Table 1, together in distinct locations of the body, namely the hypothalamus, pituitary gland, and reproductive organs. An endocrine imbalance accounts for about 10% of the major causes of infertility [35]. During stress, hormone and acoustic tests revealed increased salivary cortisol levels and related alterations in voice pitch, vocal tract resonances (formants), and speech speed. The hormones that are generated in response to stressful conditions aid a depressed body setup that has the potential to cause long-term neuroendocrine changes that impact female fertility [10]. During the normal estrogen cycle, at the follicular phase, the hypothalamus secretes the regular standard outcome of gonadotropin-releasing hormone (GnRH); this results in the pituitary gland's pulsatile secretion of luteinizing hormone (LH) release, and stress may induce abnormal hormonal production that may mislead the estrogen cycle and end up in failure in conception [36]. Under stress conditions, the serum level of hormones alters to adapt an individual to the persisting circumstances; however, long-term exposure to stress may lead to deleterious endocrine disorders that may have a negative influence upon conception [36].

**Table 1.** Reproductive hormones and their generalized functions.

| Reproductive Hormones | Functions |
|---|---|
| Gonadotropin-Releasing Hormone | • Provokes the formation of FSH and LH hormones from the pituitary.<br>• In males, it aids in the production of testosterone from the testes.<br>• In females, it aids in the production of estrogen and progesterone from ovaries. |
| Follicle-stimulating Hormone | • In males, it stimulates testicular growth.<br>• In females, it stimulates egg growth in the ovaries. |
| Luteinizing Hormone | • In males, it stimulates the testes to produce testosterone.<br>• In females, it stimulates the ovaries to produce estrogen. |
| Prolactin | • In males, it increases the production of LH hormones causing testosterone secretion that results in enhanced spermatogenesis.<br>• In females, it provokes breast growth and stimulates milk production post-pregnancy. |
| Inhibin | • In males, negative feedback controls GnRH secretion from the pituitary.<br>• In females, it regulates the FSH secretion from the pituitary. |

**Table 1.** *Cont.*

| Reproductive Hormones | Functions |
| --- | --- |
| Testosterone | • In males, it monitors the growth as well as the development of the reproductive system. It stimulates the growth of mature sperm.<br>• In females, it combines with estrogen and aids in the maintenance and damage repair of the female reproductive system. |
| Estrogen | • In males, it regulates libido, bone, muscle mass, and the production of sperm.<br>• In females, it maintains breast health, reproductive ability, and female characteristics. |
| Progesterone | • In a male, it contributes to sperm production.<br>• In the female reproductive system, it aids sperm capacitation. |
| Thyroid Hormone | • In males, it enhances the basal metabolic rate.<br>• In females, it regulates metabolism and contributes to reproductive tissue development. |
| Anti Mullerian Hormone | • In males, it obstructs the development of female characteristics.<br>• In females, it enhances the ability to get conceive. |

*3.1. GnRh*

In stressed conditions, the gonadotrophin-secreting hormone, which is the primary driver of reproductive function, is found in a reduced amount [37]. Neurons that are responsible for the synthesis of GnRH have a neural control point for modulating the reproductive function in vertebrates, and those neurons also highly represent the stress responses on reproduction since they are capable of monitoring the physiological status of the body [36]. Stress normally triggers the activation of the hypothalamus-pituitary-adrenal (HPA) axis as well as inhibits the hypothalamus-pituitary-gonadal (HPG) axis which is dealing with the secretion of GnRH. The stress may also interfere at any level of the HPG axis and may influence the reproductive function [38]. Impairment in gonadotropin production and its action results in a spectrum of changes such as relative or absolute LH and FSH deficit, which affects the gametogenesis process and steroid synthesis in gonads, lowering fertility and estradiol levels [37].

*3.2. Inhibin*

Inhibin is a gonadal hormone that exhibits a negative feedback mechanism on the FSH secretion which was excreted by the gonadotropin cells in the pituitary gland [37]. Following the separation of inhibin from porcine follicular fluid in 1986, activins were recognized as ovarian hormones, which increase pituitary gland FSH secretion. Activin A, in particular, was shown to be the isoform with the greatest physiological value in humans. The present understanding of activin A extends beyond the reproductive system, allowing it to be classified as a hormone, a growth factor, and a cytokine [39]. Various reports suggest

that inhibin B is a potential biomarker for determining the oxidative stress in Sertoli cells which are the major production site of inhibin B [40].

### 3.3. Testosterone

Testosterone is an androgen that performs an essential function in the growth, reproduction, and maintenance of a healthy body. Sperm formation (spermatogenesis) is regulated by both endocrine and paracrine systems. FSH and LH are both involved in the endocrine stimulation of spermatogenesis and then work through the intermediate testosterone generated by the Leydig cells of the testis [41]. There are only a few reports which insist on the effect of stress on testosterone synthesis and male infertility. Psychological stress primarily reduces the serum testosterone level, followed by the secondary elicit of serum LH and FSH levels, which may have an impact on the semen quality by reducing the rate of motility, count, and normally functioning spermatozoa [42]. In accordance with the guidelines of the Endocrine Society, T replacement therapy is the standard treatment for men who suffer from symptomatic hypogonadism [43]. Intratesticular testosterone is also an important factor for the spermatogenesis process, virility, and male fertility. Testicular biopsy is the only way to measure intratesticular testosterone concentration, which is invasive and can lead to many complications [44]. A transdermal testosterone pretreatment may develop ovarian sensitivity for FSH levels. As well, in low-responder in vitro fertilization (IVF) individuals, gonadotropin therapy had increased follicular response. This perspective leads to higher levels of gonadotropin and increased follicular response compared to the mini-dose GnRH agonist protocol [45]. Few other research studies had reported that after 21 days of unintended chronic stress in adult male rats, reduced body weight, serum testosterone levels, and genital index were observed [37].

### 3.4. FSH

In both males and females, a dimeric glycoprotein gonadotropin hormone called FSH and its receptor play an essential function in the generation of follicles as well as in regulating the process of steroidogenesis in the ovary and spermatogenesis in the testis. It is generated by the anterior portion of the pituitary gland in various isoforms physiologically and targets the cells of gonads [27,46]. It coordinates its function with LH through G protein-coupled receptors (GPCRs) and alters steroidogenesis and cellular metabolism to regulate reproduction [47]. The interrupted secretion of these hormones by the pituitary glands will lead to infertility and testicular functional disruption. Furthermore, this gonadotropin deficiency stands as the major risk factor accounting for about 0.5% of men's infertility. FSH interacts with its receptor on the Sertoli cells, triggering those cells in adults to secrete regulatory nutrients that are essential for germ cell maturation [47,48]. Furthermore, the circulating concentration of the germ cells directly depends on the volume of Sertoli cells [49]. It has been extensively employed in assisted reproductive technologies (ART) [50]. This hormone establishes its functionality upon binding to the receptors located in granulosa and Sertoli cells. Any polymorphism in the Follicle Stimulating Hormone Receptor gene figures out infertility in women. It may also lead to ovarian failure and primary or secondary amenorrhea [51]. The level of serum FSH has a significant correlation with stress. During psychologically stressful conditions, serum FSH was observed to lower, which may influence sperm motility and functioning [42].

### 3.5. LH

The production of LH is the cornerstone of the effective spermatogenesis process, secondary sexual characteristics, functions, and psychoactive and biosynthetic (anabolic) actions. Abnormalities in the function of LH can affect spermatogenesis, leading to infertility [49]. In men, in cases of infertility due to hypogonadotropic hypogonadism, sperm can be restored with HCG or human menopausal gonadotropins (hMG). The higher range of *body mass index* (*BMI*) will alter the concentrations of seminal plasma and male reproductive hormones, enhance oxidative stress, and affect sperm quality. These consequences may be

linked with male infertility [52]. Longer duration of infertility, women's age, and weight gain increase stress hormone levels, which exhibit decreased antioxidant activity that may lead to infertility [46]. For the first time, Gamma-aminobutyric acid (GABA) signaling in the MePD suppresses pulsatile LH secretion induced by psychological stress and suggests a functionally important MePD GABAergic projection to the hypothalamic GnRH pulse generator [53]. Chronic stress reduces kisspeptin content in the Anteroventral Periventricular Nucleus fetus and GnRH in the preoptic area in females and disrupts the LH surge, thereby disrupting the estrous cycle and fertility, leading to reduced pregnancy and embryo numbers [54]. p62 deficiency provides a new pathway, the GnRH-p62-OXPHOS (Ndufa2)-$Ca^{2+}$/ATP-LH pathway, which on a theoretical basis exhibits the abnormal secretion of pituitary LH through OXPHOS signaling of mitochondria that may be a risk pathway for causing infertility in females [55].

### 3.6. Prolactin

Prolactin is a polypeptide hormone, an essential factor for male reproduction secreted by lactotrophs [56]. Treating prolactin elevation causes an affirmative metabolic effect on the patient, namely controlling glycemic content and other metabolic parameters [57]. Men experiencing long-term psychological stress have greater levels of prolactin which are not noticed in women [58]. Infertile men with oligo- and azoospermia, impaired movement, and hypogonadotropic hypogonadism have an extensively increased volume of serum prolactin levels as well as levels of seminal plasma. High prolactin in serum exhibits a chronic impact on the process of spermatogenesis in infertile men [59].

### 3.7. Estrogen

It has been debated for a long time that males also have the estrogen hormone. According to early studies, estrogen was harmful to male reproduction since the exogenous treatments led to abnormal developmental outcomes. Estrogen synthesis in the testis and elevated levels of 17-oestradiol in the testis fluid are recommended as an essential factor in healthy male reproduction, but they are still a controversy [60]. Based on the species, the receptors of estrogen such as estrogen receptors (ERs), ESR1 (ER-alpha), and ESR2 (ER-beta) are expressed in certain cells of the testis and also in the epididymal epithelium. Through the enzyme named aromatase, the testis secretes a crucial amount of this hormone [61]. In the human body, biologically active aromatase is expressed by Leydig cells, immature germ cells, and in spermatozoa. Additionally, ESR1 and ESR2 are found in spermatozoa and germ cells, and it is important to note that spermatozoa contain a truncated form of ESR1. These observations unmistakably imply that estrogens are probably involved at various stages of the development of male germ cells [62]. Historically, testosterone and estrogen were thought to be male and female sex hormones, respectively. However, estradiol, the most common type of estrogen, is also important in male sexual function. It is required in males to regulate libido, erectile function, and spermatogenesis [48]. Estrogen is crucial for different physiological functions, including controlling energy metabolism, mineral balance, stress responses, and sexual development [63]. It is a powerful vasodilator that increases blood flow in several organs throughout the body, with reproductive tissues, particularly in the uterus, experiencing the greatest effects [64].

### 3.8. Progesterone

Progesterone is the steroid hormone that plays a vital role in female reproductive function. It helps the mature ovary to get released into the uterus. During pregnancy, this hormone facilitates implantation and suppresses myometrial contraction to promote uterine growth. Physiological effects of progesterone hormone are particularly arbitrated by an intracellular progesterone receptor (PRs). The PR protein consists of a central globular DNA-binding domain (DBD), a C-terminal ligand-binding domain (LBD), and an amino-terminal domain (NTD) that is largely composed of intrinsically disordered (ID) protein [65]. This PR protein expressed in two isoforms—progesterone receptor-A

and progesterone receptor-B—was identified in the 1970s [66]. The AF1 and AF2 are the two functional transcriptional domains located at PR. AF1 is found in NTD, and AF2 is found in LBD [65]. The expression of progesterone receptor-A is essential for the ovulatory function in response to progesterone, and in most cases, progesterone receptor-A inhibits the progesterone receptor-B. The PR-B is a strong co-activator that regulates the target gene response to progesterone [67]. The impaired PR endometrial expression during implantation leads to infertility [68]. The primary hormone during the luteal phase is progesterone. It is essential in both orchestrating the uterus for a prospective gestation and sustaining it. Because there is frequently a luteal phase shortfall in assisted reproduction therapies, it is vital to supplement this key period to get the greatest outcomes, not only for implantation but also for sustaining the pregnancy [69].

### 3.9. Anti-Mullerian Hormone (AMH)

The Anti-Mullerian hormone (AMH) is made of small developing follicles in the ovary produced by granulose cells [70]. It is a homodimer glycoprotein involved in transforming the growth factor-B (TGFB) super family [71] and is an outcome marker of the ovarian reserve pool belonging to chromosome 19 short arm [72]. It acts as momentum for the quantitative aspect of ovarian reserve constitutes by the quality and quantity of primordial follicles [73]. The primordial follicles undergo atresia by FSH after puberty when the HPG endocrine axis is activated [74]. It produces 140 kDa disulfide-linked homodimer covalently bonded as a monomer of 560 amino acids. AMH exists as an active form of C-terminal about 25 kDa dimer and N-terminal pro region about 110 kDa. It is known for its serum marker assessment of spectrum ovarian reserve and PCOS [75]. AMH provides two distinct receptors, AMHRI and AMHRII, involved in signal transmission relating Suppressor of Mothers Against Decapentaplegic (SMAD) proteins [73]. These receptors transduce intracellular signals effectively through nuclear and physiological binding to AMH, which exhibits an essential function of prenatal gonadal sex differentiation by developing mullerian ducts in males. During embryonic development by the eighth post-conception week, the AMH hormone is produced by immature Sertoli cells in the testes until the age of 2 years post-natal, which declines over pubertal years; then, ultimately, it becomes untraceable in adults due to high concentrations of testosterone. The lower devotion of AMH level in males leads to the growth of the male and female genitalia. In females, analyzing serum AMH levels with autoimmune disease associated with the possibility of infertility [72].

### 3.10. Prolactin

A polypeptide hormone prolactin is produced by the lactotrophs, a specialized cell of the anterior pituitary gland. Other than lactation during reproduction, it plays an important role in many homeostatic roles [56]. Transmembrane prolactin receptor (PRL-R)is involved in prolactin signaling which is expressed in a wide range of tissues [75]. The hormone prolactin plays an essential function in determining the pathogenesis of schizophrenia and related psychoses [76,77]. Prolactin is concluded as the remarkable marker for diagnosing stress [78]. During hypoglycemia, the prolactin levels are gradually raised, and symptomatic neuroglycopenia is achieved. During the period of pregnancy and lactation, a notable elevation in brain PRL-R expression has been expressed as well as a rise in prolactin gene expression in the hypothalamic region [79]. The adenohypophysis hormone prolactin facilitates adaptive behavior, induces analgesia, and enhances grooming behavior. It plays a protective role against biological modifications induced by stress. There might be a chance of a higher magnitude of increase in prolactin in women than in men, and this is dependent on estradiol levels [79].

### 3.11. Thyroid Hormones

The frequency of thyroid disease occurrence in females of fertile age has been categorized to be thyroid autoimmunity, hypothyroidism, and hyperthyroidism, wherein

infertility is quite common in women with thyroid dysfunction relating to Graves and Hashimoto's disease [80]. The association of thyroid-stimulating hormone (TSH) elevation and thyroid autoimmunity (TAI) links infertility with a low ovarian reserve pool [81]. The severity of thyroid dysfunction leads to menstrual disorders through the interactions with the hypothalamic-pituitary-ovarian axis by direct and indirect means [82]. The probability for the presence of TAI leads to infertility either by direct or indirect reflections of immune imbalance [83]. A female associated with infertility related to thyroid disorder is then first treated to normalize thyroid dysfunction, and the workout of infertility is resolved by ART treatment [84,85]. Thyroid hormone receptors are directly identified on the testis, located on Sertoli cells in seminiferous tubules [61,86]. Sertoli cells are known to be the first somatic cells in the testes to help in supporting and nurturing spermatogenesis where TR plays a vital role in mediating the thyroid hormone in sperm production [87]. Thyroid dysfunction causes several alterations in semen quality including the volume, density, motility, and morphology of sperm [39]. The elevation of thyroid dysfunction in the condition of hyperthyroidism creates a short period of proliferation in Sertoli cells and leads to a delay in spermatogenesis [88]. Hypothyroidism and hyperthyroidism changes spermatic production and characteristics by creating alterations in thyroid receptors, thus thyroid hormone imbalance may be involved in causative mechanism of male infertility [89]. Thyroid hormones (T3 and T4) are necessary for the healthy state and operation of the female conceptive system because they regulate ovarian, uterine, and placental tissue metabolism and development. As a result, in women as well as in animals, both hypo- and hyperthyroidism can cause sub-fertility or infertility. Menstrual irregularity, anovulation, premature birth, preeclampsia, restriction in the growth of intrauterine, postpartum thyroiditis, and mental impairment in children are all well-documented consequences of maternal thyroid dysfunction [48].

## 4. Various Biological Mechanisms Related to Stress That Leads to Infertility

Stress has a negative impact on reproduction; nevertheless, the mechanism behind this is not well understood. Cadmium build-up to generate free radicals $Cd^{2+}$ is a complex II non-competitive inhibitor, resulting in semi-ubiquinone reduction which is reactive and unstable, and it may produce necrosis and apoptosis by inducing a long-term production of ROS [90]. This may also result in infertility or poor pregnancy outcomes due to high levels of ROS and poor antioxidants, likely glutathione, that affect the entire life of men and women [10,91]. Through the mechanisms of alpha-amylase activity, using amylase salivary concentrations via the sympathetic-adreno-medullary (SAM) axis reduces fertility by interacting with catecholamine receptors, changing blood flow via the fallopian tubes and gamete transfer [92]. DNA damage caused by oxidative stress is a major cause of sperm dysfunction, leading to congenital malformations and complex neuropsychiatric disorders [93]. Due to low antioxidants, spermatozoa are sensitive to DNA damage. ROS-mediated sperm destruction is key for sperm mobility and the embryo's paternal genetic contribution, which creates a link between seminal leukocytes and ROS generation. Thus, ROS is inversely connected with sperm DNA integrity. Superoxide dismutase (SOD) is an enzymatic antioxidant, which protects against oxidative stress, a relationship between low seminal catalase activity and male infertility, and the other antioxidant glutathione peroxidase (GPX) with mercapto-succinate causes a significant increase in sperm lipid peroxidation [94]. The inflammatory response deposition under Th1-lymphocyte and M1-macrophage responses regulates cytokines, adipokines, and myokines which are obesity-related inflammatory responses that disrupt the hypothalamic-pituitary-gonadal (HPG) axis preventing the release of hypothalamic GnRH and subsequent release of FSH and LH [95] that might cause intracellular oxidative damage to spermatozoa via increased lipid peroxidation. Thus, the excess formation of ROS affects reproduction [96]. Cortisol, a stress hormone, may inhibit estradiol-17 production in the ovary, resulting in amenorrhea, anovulation, and menstrual abnormalities as a result of stress [97]. Human and animal reproductive activities are primarily governed by the HPG axis as illustrated in Figure 1,

and reduced GnRH action in both males and females, resulting in lower gonadotropin levels operating stress-related variables [98]. h various fertility medicines, however, is just Nutritionally induced oxidative stress consists of excessive macronutrient intakes that might cause oxidative stress that may favor inflammation via NF-B-driven cell signaling pathways [99] and make a response to several consequences such as nutritional deficiency, viral infection, and genotoxic stress [100]. The equilibrium shifts towards an excess of ROS, and oxidative stress occurs, which can cause oocyte aging and reduce oocyte number and quality [101]. It can also cause embryo fragmentation and the emergence of a variety of developmental defects and is thought to be one of the major causes of spontaneous and recurrent miscarriage [102]. Reproductive illness due to ROS leads to polycystic ovary syndrome (PCOS), endometriosis, and pre-eclampsia [103]. Pre-eclampsia is linked to fetal morbidity and death, monitored using a valid indicating marker, F2-isoprostanes [104,105]. Endometriosis and pre-eclampsia can impair embryo implantation by stimulating the surge of reactive oxygen and nitrogen species [106]. The microbiota is referred to as a group of microorganisms for the physiological process of the host present in mucosal tissues of the gut, reproductive tract, and skin [107]. The microbiota is very important for sustaining the mucosal barrier structural integrity, pathogen refuge, and immunomodulation [108]. The convulsion in microbiota leads to a decrease in the ratio of pathogenic organisms and a reduction in entire microbial diversity [109]. The gut and intestinal microbiota may affect male and female fertile capability through the impact on inflammatory conditions [110]. Oxidation of glyceraldehyde-3-phosphate dehydrogenase (GAPDH-S) resulting in H2O2 generation has been reported to reduce sperm motility and inhibit GAPDH-S activity [111].

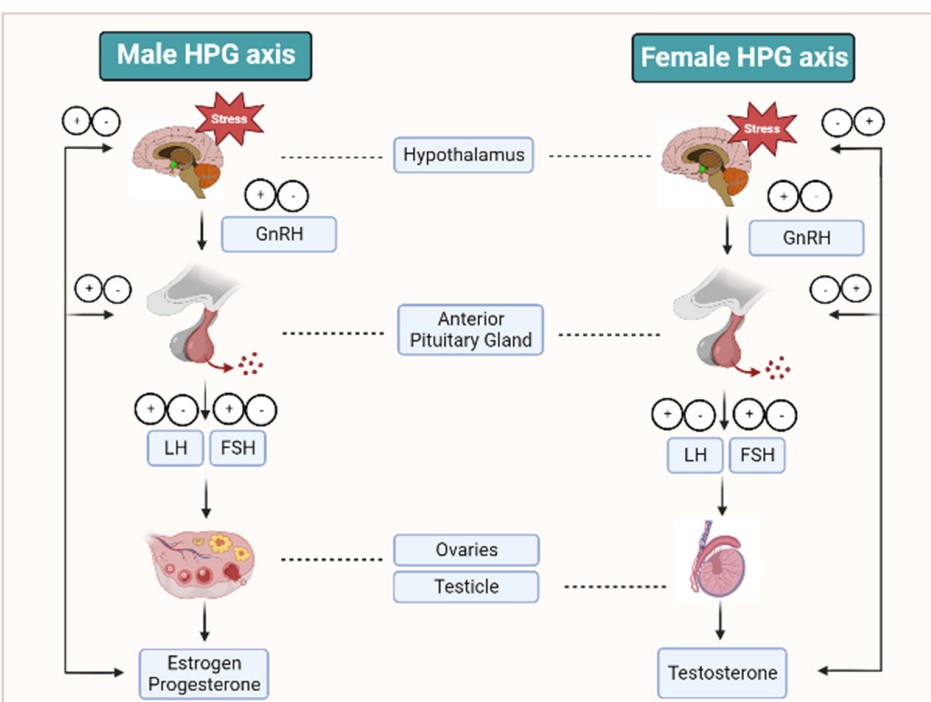

**Figure 1.** HPG axis and the reproductive hormones under stress conditions. This figure illustrates that at prolonged stress conditions, the HPG axis produces increased or decreased quantities of hormones that lead to hormonal imbalance, which is associated with the major risk factor PCOS, and this ultimately leads to infertility.

## 5. Stress and Other Reproductive Ailments

Varicocele, metabolic syndrome, sexually transmitted diseases, tobacco use, alcohol, bacterial prostatitis, microbial infections, mutations, and viral infections are all potential causes of OS and inflammation [112]. Furthermore, increases in sperm quality following varicocelectomy may not always result in spontaneous conception [113]. Stress factors

and their relative mechanisms may also cause immune dysregulation in reproductive health [114]. Physical and emotional stress may lead to pubescence, and, at the reproductive stage, they may cause impaired oogenesis and spermatogenesis resulting in transitory sterility in women and permanent infertility in males, respectively. Corticotrophin-releasing hormone (CRH) found in both male and female reproductive systems serve as an autocrine and paracrine modulator that has participated in the regulation of steroidogenesis and the inflammatory processes of the ovary, the endometrium, and placental CRH [115]. Abnormal CRH may cause various conditions including preeclampsia, endometrial growth retardation, abnormal placenta invasion, and preterm delivery [116]. Long-term inhibition of the metabolism by increased activation of glucocorticoid secretion in stress leads to persistent reproductive dysfunction [117]. Melatonin, an important factor of reproduction, attenuates and counteracts oxidative stress. It acts as a scavenger for ROS [118]. During stress, a variety of components of the HPA axis can suppress the activity of the HPG axis at distinct multiple levels. The HPG axis is regulated by a hypothalamic neuropeptide, gonadotropin-inhibitory hormone (GnIH), which is the key negative regulator of reproduction [119].

PCOS—Almost 7 to 15% of women belonging to the reproductive age are diagnosed with a metabolic gyne-endocrine disorder named PCOS [120,121]. It results in clinical hyperandrogenism, biological dysovulation, and infertility [122]. It is also associated with certain pathophysiological symptoms including glucose intolerance, diabetes mellitus, atherogenic dyslipidemia, systemic inflammation, non-alcoholic fatty liver disease, hypertension, coagulation disorders, severe acne, neuroendocrine alterations, hirsutism, insulin resistance, increased androgen, adiponectin levels, adiposity, menstrual irregularities, elevated endocrine, metabolic modifiers, and ovarian abnormalities [123,124]. In the case of PCOS, the OS index is high and is significant even after BMI adjustment. PCOS patients with lower BMI have lower emotional distress than obese patients [125,126]. Women with PCOS have abnormal biomarkers of OS and have lower amounts of antioxidants such as catalase and ferroxidase [127,128]. OS modifications with d-*chiro*-inositol, an isomer of inositol, vitamin D, and probiotic administration are effective beneficial strategies in reducing oxidative stress in PCOS women [129–131]. Depression is the most common effect found in patients with PCOS which has a serious effect on their quality of life [132]. It is three times more common in PCOS women than in non-PCOS women [133,134]. Depression is strongly associated with insulin resistance which serves as a physiological mediator [135]. Electro acupuncture is also a method of treatment provided to PCOS patients. It improves the symptoms of depression and people's quality of life [136].

Endometriosis—A pathological condition that exhibits the presence of endometrial-like tissue around the uterus and triggers chronic inflammatory symptoms; most of the women affected are asymptomatic. About 35–50% of females affected with endometriosis are diagnosed with infertility [137]. OS is highly linked with the prognosis of endometriosis; OS and ROS production is found greater in endometriosis, and it reflects damage and proliferation. It also affects multiple physiological functions such as ovulation, implantation, oocyte maturation, luteolysis, and the maintenance of the luteal phase during pregnancy which partially reflects the infertility status [138]. Infertility caused by endometriosis can be treated by the supplementation of antioxidants such as Vitamin C and E, melatonin, etc. [139]. In endometriosis, individuals correlated with infertility have impaired secretion of gonadotropin hormones, which is proportional to the disease severity [140].

## 6. Impact of Stress on Infertility Treatment

Fortunately, major advancements in the medical and surgical treatment of infertility have lately been developed. The total increased risk associated with various fertility medicines, however, is just 1–2% [141]. PCOS is the greatest prevalent condition of anovulatory infertility. It is treated with various medications; oral ovulation induction medications are the first-line pharmacological therapy for women with the condition who are unable to conceive. Clomiphene citrate is the most well-established therapy, while aromatase

inhibitors have shown promise, although efficacy has yet to be verified in randomized controlled trials [142]. Repeated clomiphene medication may be linked to an increased incidence of hypospadias and neural tube abnormality [143]. Patients with PCOS who were clomiphene-resistant showed lower antioxidant levels than the people who were clomiphene sensitive but did not have PCOS. Myeloperoxidase levels followed the same pattern, which might be explained by a compensatory mechanism [127]. Females with PCOS have increased levels of inflammatory markers as well as lower levels of antioxidants. N-acetylcysteine appears to be effective in restoring oxidative equilibrium and fostering high-quality oocytes with optimal follicular size. It also boosts ovulation and conception rates while having fewer negative effects including ovarian hyperstimulation syndrome, which is frequent with other drugs [144]. Premature ovarian insufficiency (POI) affects at least 1% of all women and creates long-term health issues as well as psychological stress. POI-induced infertility was once thought to be absolute, with infertility therapy having little or no efficacy. In general, it has been assumed that medicine can only give little assistance to these individuals [141].

## 7. Factors That Cause Decreased Infertility Rate

In these modern days, infertility has become the most prevalent health issue, and it has been impacted by various risk elements including lifestyle habits, diet, environment, stress, and endocrine disruptors. The great attention to quality-of-life choices such as food habits, stress, drinking, smoking, and obesity affects the physiology of an individual and exerts long-term effects [145]. Healthy lifestyle behaviors from preconception through postpartum are regarded as an important precaution for attaining healthy pregnancies and avoiding gestational disorders. Healthy diet control, BMI control, physical exercise, and physical, mental, and emotional health are among the World Health Organization's (WHO) preconception priorities [146].

### 7.1. Nutrition

A healthy diet is opulent in carbs, fiber, fruits, and vegetables, and folate is linked to enhanced sperm quality [147]. Another possible benefit might be antioxidants, which are majorly essential for sperm capacitation and perform a crucial in ROS scavenging activity, where ROS overloading might cause sperm malfunction and affect its motility, damage DNA, and lower its membrane integrity. The antioxidants aid in the removal of excess ROS or convert ROS into molecules that are less harmful to cells. It is recommended to have a diet rich in antioxidants [148].

### 7.2. Smoking

Usage of tobacco causes exposure to about 4700 mutagenic substances such as hazardous chemicals and polycyclic aromatic hydrocarbons which lead to oxidative stress in the testes that affects spermatogenesis and steroidogenesis and results in poor chromatin condensation, DNA integrity, oocyte binding, and epigenetic alterations [149].

### 7.3. Caffeine Consumption

Caffeine usage is associated with elevated levels of testosterone as well as the sex hormone binding globulin (SHBG). It alters the crucial properties of Sertoli cells which include the glycolytic and oxidative properties that may impair reproductive ability in men. In embryos as well as in adults, caffeine can modify the hormonal system and has a modifying influence on the germinative epithelium [150].

### 7.4. Gadgets

The increased use of mobile phones, as well as the storing of phones in trouser pockets, has been identified as a source of harmful radiation to the male reproductive system. A recent study found that DNA fragmentation was the only variable that changed among mobile phone users with high usage (>4 h daily) who kept their phone in their trouser

pocket [151]. The growing usage of hands-free kits with belt-holstered phones, in particular, has generated concerns about the possibility of radiofrequency-electromagnetic radiation (RF-EMR) exposure to the gonads [152].

### 7.5. Age

Age has a high impact on human reproduction. It also affects the HPG axis function, resulting in alterations in other reproductive hormones. This is because males go through a lot of physiological changes. As early as age 35, sperm parameters begin to drop steadily. The volume and motility of the sperm decline, and the morphology may become progressively aberrant [153].

### 7.6. Marijuana

Given the deleterious effects of marijuana on male reproductive physiology, marijuana includes the cannabinoid delta-9-tetrahydrocannabinol (THC), which inhibits the secretion of hypothalamic luteinizing hormone-releasing hormone (LHRH)and adenohypophysis LH. Furthermore, 1 THC's central blockage of the HPG axis in males causes a dose-dependent decrease in plasma T levels, which may take up to 3 months to normalize following discontinuation. Chronic usage can cause gynecomastia, decreased libido, erectile dysfunction (ED), and ejaculatory dysfunction [152].

### 7.7. Obesity

Obesity was related to the risk for erectile dysfunction (ED). Being overweight or obesity is observed in 79% of males reporting ED symptoms. In very obese males, BMI was related to higher avoidance of sexual contact and increased difficulty with sexual performance, resulting in decreased satisfaction with sexual life [154]. This leading risk factor influences the metabolic processes; namely hypercaloric diet alters the low energy state which targets the HPG axis, thus creating a negative impact on female fecundity [155].

### 7.8. Stress

Stress is referred to as the non-peculiar response of the body to danger or unexpected events as well as the ability to trigger various stimuli [156]. Stress may be due to sexual attitude, marital life, quality of life, work life, motherhood, social tension, etc. Any of these may have a detrimental influence on sperm quality. The recent scenarios that may lead to stress are illustrated in Figure 2 [157]. Stress may not have a great impact on sperm morphology, but it has a 39% reduction in sperm quantity and a 48% reduction in motility. Thus it is indicated that stress is negatively associated with semen quality [150,152].

### 7.9. Anxiety

As an outcome of infertility treatment, the individuals affected by infertility have reported sexual distress, anxiety, frustration, emotional distress, marital problems, and loss of self-esteem [158]. The anxiety level is high in infertile men and women compared to a normal fertile control group that has been recently founded in research. Most of the psychological consequences show that women with infertility have a stringent experience of anxiety, and the women undergoing infertility treatment have to deal with two different types of stressors: the loss of parenthood hopes and the threat of certain infertility lead to a chronic stressor, and the fertility treatment itself leads to an acute stressor [159,160]. The cost of infertility treatment causes financial pressure, and the unpredictability of this treatment influences the sexual relationship [161].

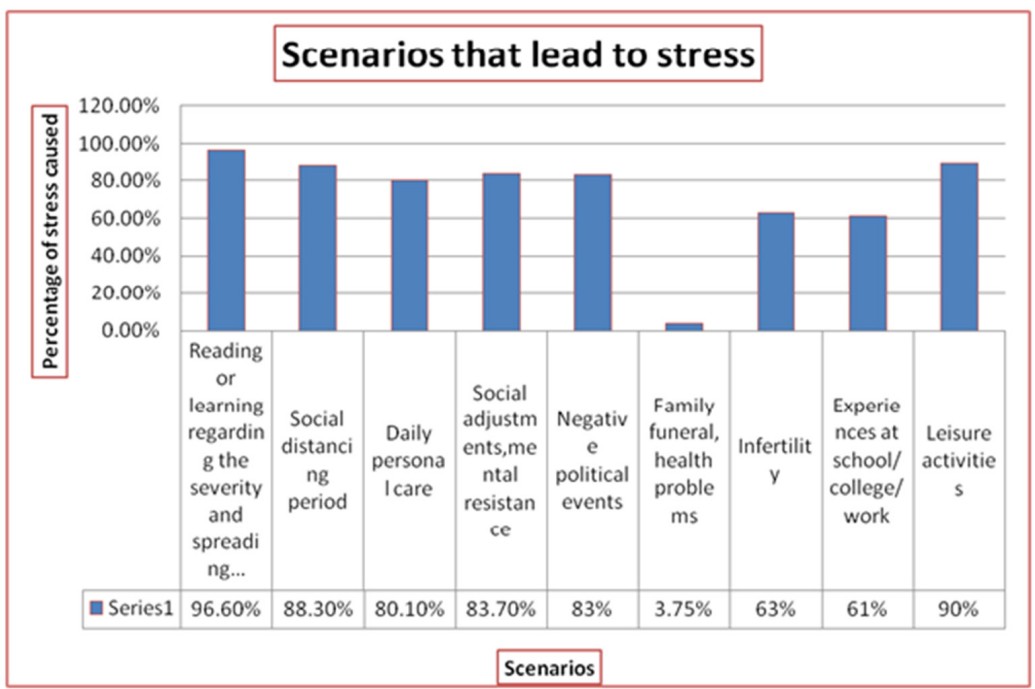

**Figure 2.** Scenarios that lead to stress. This figure illustrates certain scenarios in human life that may lead to acute or chronic stress conditions and depicts them as a graphical representation of the percentage of stress they induce.

*7.10. Depression*

In the case of women, culture, clinic visits, and sexual and social concerns were the elements associated with the symptoms of depression, whereas in men, financial status and social pressure were linked with depression symptoms [158]. Almost 30% of the population is suffering from clinical depression which causes lower libido, loss of interest, lower live births, etc. Thus, it stands as the major factor influencing the success rate of pregnancy, and its treatment deserves particular attention [159].

**8. Infertility Management**

*8.1. Supplementations*

Various formulations and uses of oral antioxidants that are readily available in the market were expected to treat infertile men and also increase the pregnancy outcomes in women with multiple miscarriages, but in certain case reports, it has been registered that over-usage of antioxidants will have an impact on sperm cells [162,163]. Another systematic review identified antioxidant vitamin E, vitamin C, N-acetylcysteine, selenium, and zinc as significantly beneficial for treating men with OS-related infertility [164].

*8.2. Health Management*

Exposure to harmful toxins or chemicals, pollution, and heavy metals; often taking hot baths; prolonged sitting or driving; and a workplace without great ventilation may induce OS. Hence, carrying out healthy habitual activities such as wearing personnel protective equipment, respiratory protection, eye protection, and hazard banding; following yoga and meditation; regular exercise; adequate sleep; and balanced nutrition are recommended [9,33,94,164]. Anxiety, depression, and fertility-related quality of life illustrate an improvement with participation in a yoga program [165]. Management of mental health during the pregnancy period and postpartum has to involve the obstetrician, pediatrician, and concerned members of the family [166]. Regular exercise leads to improved sperm and semen quality in lifestyle-induced infertility by increasing testicular antioxidant de-

fenses, reducing levels of pro-inflammatory cytokines, and enhancing the steroidogenesis process [167].

*8.3. Therapeutics*

On understanding, the etiopathology mechanisms facilitate clinicians to undergo optimal treatment to overcome infertility. For most cases of infertility, targeted therapy that is a contributory cause of infertility is well known. Forthis, either hormonal or non-hormonal assistance is followed. Whereas idiopathic infertility is treated with empirical non-hormonal therapeutic assistance [168,169], the purpose of hormone therapy is well-defined in men with pin-down abnormalities which include hypogonadism [15]. A combined treatment of surgical approaches and hormonal therapy or customized therapeutic strategies have reported convincing results in recent years [168]. The therapeutic elements may include antioxidants, antibiotics, anti-inflammatory drugs, micronutrients, and non-hormonal compounds [169].

*8.4. Psychological Support*

The psychological status of infertility patients should be assessed ideally by a mental health professional with the framework of a questionnaire. This may hopefully increase the pregnancy outcomes. The support may be extended by providing the individuals with information on self-care skills and relaxation techniques to resolve anxiety or any distress conditions [33,169].

### 9. Conclusions

The couple facing infertility will be going through a stressful and depressing period of life. Silent struggle with infertility is prevailingly common among today's individuals. The diagnosis and treatment are a tremendous burden. The impact of stress on infertility is still a controversial concept, However, stress, anxiety, and depression are found to have interventions in the reproductive rate. This review emphasizes that stress can be a major contributor to infertility, and it may also affect the treatment success rates, concerns about fertility, and the role of hormones in fertility outcomes. Finally, stress may be one of the reasons for infertility in this modern era. Furthermore, the documentation on the hormonal pathways that correlate the stress and infertility should be emphasized, which could enhance our view on the association between the common symptoms, stress, and infertility.

**Author Contributions:** Conceptualization, Methodology, Validation, and Draft preparation, S.R., M.D., P.P. (Prasad Poornima), S.E., S.S. and A.V.A.; Reviewing and editing, S.R. and A.V.A.; Formal analysis, A.J., I.P.G., V.B., P.P. (Palanisamy Priyanga) and J.V.; Language correction, S.R., B.B. and A.V.A. All authors have read and agreed to the published version of the manuscript.

**Funding:** This research received no external funding.

**Data Availability Statement:** The authors confirm that the data supporting the review are available upon reasonable request from the corresponding author.

**Conflicts of Interest:** The authors declare no conflict of interest.

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
