# Peer review of "Role of Hormones and the Potential Impact of Multiple Stresses on Infertility"

_stresses, doi:10.3390/stresses3020033_

Round 1

Reviewer 1 Report

A thorough language edit is needed to make this contribution readable

Author Response

Reviewer Comment: A thorough language edit is needed to make this contribution readable.

Authors reply: Thank you for your valuable comments. As per the suggestion, the language edit is done.

Reviewer 2 Report

Dear Editor,

Thank you for giving me the opportunity of reviewing manuscript entitles “A COMPREHENSIVE REVIEW OF THE CRUCIAL ROLE OF SEX HORMONES AND POTENTIAL IMPACT OF PSYCHOLOGICAL STRESS ON INFERTILITY” in  Journal Of stress. This manuscript is well-written and have not  sufficient novelty.

The introduction is not well-written.

There is no information about methodology of this review.

what is differences between this review and other published reviews?

In sub-heading 5 why authors discuss about thyroid hormones? It is better separate it.

The discussion about male hormones and psychological issues is not well-written. I mean, it is mainly describe the hormone secretion and mechanisms.

what is the relationship between risk of CVD and objective of this study.

instead of PCOD, you should use PCOS.

Age is not a lifestyle factor.

Author Response

Reviewer Comment: Thank you for allowing me to review the manuscript entitled “A COMPREHENSIVE REVIEW OF THE CRUCIAL ROLE OF SEX HORMONES AND POTENTIAL IMPACT OF PSYCHOLOGICAL STRESS ON INFERTILITY” in the Journal Of stress. This manuscript is well-written and does have not sufficient novelty.

Authors reply: Thank you for your valuable comments. Our manuscript’s novelty in describing the impact of stress on each reproductive hormone was not found in any of the review publications so far further other reproductive ailments related to stress and multiple mechanisms associated with stress that causes infertility is collectively documented in our manuscript.

Reviewer Comment: The introduction is not well-written.

Authors reply: Thank you for your valuable comments. The introduction is modified.

Reviewer Comment: There is no information about the methodology of this review.

Authors reply: Thank you for your valuable comments. International databases (PubMed, Scopus, Web of Science) and Google Scholar were searched for articles published from 2000 to February 2023. The search procedure was performed in English using keywords such as “stress”,” depression”,” infertility”, and “reproductive hormones”. The articles were evaluated in terms of their titles, abstracts, and full texts.

Reviewer Comment: What are the differences between this review and other published reviews?

Authors reply: Thank you for your valuable comments. Our manuscript’s novelty in describing the impact of stress on each reproductive hormone was not found in any of the review publications so far further other reproductive ailments related to stress and multiple mechanisms associated with stress that causes infertility is collectively documented in our manuscript.

Reviewer Comment: In sub-heading 5 why do authors discuss thyroid hormones? It is better to separate it.

Thank Authors reply: you for your valuable comments. The content of thyroid hormones was separated.

Reviewer Comment: The discussion about male hormones and psychological issues is not well-written. I mean, it is mainly describing hormone secretion and mechanisms.

Authors reply: The impact of stress on each reproductive hormone is shortly described.

Reviewer Comment: What is the relationship between the risk of CVD and the objective of this study? 

Authors reply: Thank you for your valuable comments. The previous axillary factors of stress were mentioned as CVD. Now the content  was removed based on overall alterations in the review

Reviewer Comment: Instead of PCOD, you should use PCOS.

Authors reply: Thank you for your valuable comments. PCOS and endometriosis are considered as the most causative factors of infertility

Reviewer Comment: Age is not a lifestyle factor.

Authors reply: Thank you for your valuable comments. Yes, based on your comments the content is modified.

Reviewer 3 Report

Ramya et al. presented the impact of stress on reproductive rates and the implications of sex hormones on infertility. This is an interesting review paper, indicating that stress can be a major contributor to infertility and it may also affect the treatment success rate, and the role of hormones in fertility outcomes. There are, however, some issues to be addressed to further improve the manuscript.

1.     Even though the authors employed the phrase  “potential impact of psychological stress on infertility”, there is no independent section for discussing the relationship between psychological stress and infertility. The authors rather discussed a lot about other types of stress such as physical stress, chemical stress, and so on. This sounds curious.

2.     Early-life stresses including adverse experiences are an urgent and major today’s problem . Effects of these issues on infertility should be also discussed.

3.     English editing should be done again.

Author Response

Reviewer Comment: Ramya et al. presented the impact of stress on reproductive rates and the implications of sex hormones on infertility. This is an interesting review paper, indicating that stress can be a major contributor to infertility and it may also affect the treatment success rate, and the role of hormones in fertility outcomes. There are, however, some issues to be addressed to further improve the manuscript.

Even though the authors employed the phrase  “potential impact of psychological stress on infertility”, there is no independent section for discussing the relationship between psychological stress and infertility. The authors rather discussed a lot about other types of stress such as physical stress, chemical stress, and so on. This sounds curious..

Authors reply: Thank you for your valuable comments. The independent section for all kind of stress that influence the reproductive rate is shortly described.

Reviewer Comment: Early-life stresses including adverse experiences are an urgent and major today’s problem. Effects of these issues on infertility should be also discussed.

Authors reply: Thank you for your valuable comments. All kind of possible stresses that may influence the human reproductive rate is shortly described.

Reviewer Comment: English editing should be done again.

Authors reply: Thank you for your valuable comments. Grammatical check and English editing is done.

Round 2

Reviewer 1 Report

Idem as in the previous round

Author Response

Idem as in the previous round.

Authors reply: Thank you for your valuable comments for better outcome of the manuscript.

Earlier Comments:

Ramya et al. presented the impact of stress on reproductive rates and the implications of sex hormones on infertility. This is an interesting review paper, indicating that stress can be a major contributor to infertility and it may also affect the treatment success rate, and the role of hormones in fertility outcomes. There are, however, some issues to be addressed to further improve the manuscript.

Even though the authors employed the phrase  “potential impact of psychological stress on infertility”, there is no independent section for discussing the relationship between psychological stress and infertility. The authors rather discussed a lot about other types of stress such as physical stress, chemical stress, and so on. This sounds curious.

Authors reply: Thank you for your valuable comments. The independent section for all kind of stress that influence the reproductive rate is shortly described. The major stresses which include the pschycological stress, emotional stress, preconception stress and biologically metabolic and oxidative stress were included in the pages 2-4.

Reviewer Comment: Early-life stresses including adverse experiences are an urgent and major today’s problem. Effects of these issues on infertility should be also discussed.

Authors reply: Thank you for your valuable comments. All kind of possible stresses that may influence the human reproductive rate is shortly described in the pages 2- 4 and 1-12.

Reviewer Comment: English editing should be done again.

Authors reply: Thank you for your valuable comments. We have done the grammar check throughout the manuscript.

Reviewer 3 Report

The manuscript was mostly improved.

Author Response

Reviewer Comment: The manuscript was mostly improved.

Authors reply: Thank you for your valuable and positive comments.